# On Equivariant Model Selection through the Lens of Uncertainty

**Putri A. van der Linden**[*,1]      **Alexander Timans**[1,2]      **Dharmesh Tailor**[1]      **Erik J. Bekkers**[1]

[1]Amsterdam Machine Learning Lab, University of Amsterdam
[2]UvA-Bosch Delta Lab, University of Amsterdam

## Abstract

Equivariant models leverage prior knowledge on symmetries to improve predictive performance, but misspecified architectural constraints can harm it instead. While work has explored learning or relaxing constraints, selecting among pretrained models with varying symmetry biases remains challenging. We examine this model selection task from an uncertainty-aware perspective, comparing frequentist (via Conformal Prediction), Bayesian (via the marginal likelihood), and calibration-based measures to naive error-based evaluation. We find that uncertainty metrics generally align with predictive performance, but Bayesian model evidence does so inconsistently. We attribute this to a mismatch in Bayesian and geometric notions of model complexity for the employed last-layer Laplace approximation, and discuss possible remedies. Our findings point towards the potential of uncertainty in guiding symmetry-aware model selection.

## 1 INTRODUCTION

Real-world tasks frequently exhibit geometric symmetries such as rotations or reflections, and equivariant predictive models for such settings have proven effective on problems ranging from medical imaging [Fu et al., 2023] to molecule synthesis [Atz et al., 2021, Batzner et al., 2022] and physics simulations [Brandstetter et al., 2021]. Such geometric knowledge is generally embedded as constraints on model expressivity, and misspecifying the inductive bias may harm performance [Petrache and Trivedi, 2023]. This has lead to recent work on *learning* equivariance and softening constraints, *e.g.* Romero and Lohit [2022], Moskalev et al. [2023], van der Linden et al. [2024], including from a Bayesian model selection perspective [van der Wilk et al.,

2018, van der Ouderaa et al., 2023]. Yet such relaxations also remain architecturally tied, and from a practitioner's *post-hoc* perspective it remains unclear what model to favor when faced with a range of pretrained options, from fully unconstrained to strictly equivariant. While a simple hold-out error comparison (*e.g.* on accuracy) is possible, measures that incorporate notions of *uncertainty* have been advocated for more robust model assessment [Begoli et al., 2019, Makridakis and Bakas, 2016], resulting in much work on uncertainty for neural networks [Gawlikowski et al., 2023].

Taking this perspective, we investigate equivariant model selection through the lens of uncertainty. Given differently constrained architectures, we verify how *post-hoc* frequentist (via Conformal Prediction intervals), Bayesian (via a Laplace-based marginal likelihood approximation) and calibration measures compare to a simple error-based evaluation. Our experiments on object shapes (ModelNet40) and molecule data (QM9) suggest that uncertainty-based model recommendations generally align with predictive error, but the Bayesian model selection framework does so inconsistently in our *post-hoc* setting. We posit that this stems from mismatches between geometric biases present in the feature representation and the approximation's limited sensitivity. Overall, our findings suggest the potential of uncertainty-aware frameworks in guiding equivariant model selection.

## 2 BACKGROUND

We next provide some brief background on equivariance and uncertainty-related topics relevant for this work.

**Equivariance and invariance.** A map $f : \mathcal{X} \to \mathcal{Y}$ is said to be *invariant* to a given transformation group $G$ if it satisfies the condition

$$f(\mathbf{x}) = f(g \circ \mathbf{x}) \quad \forall g \in G, \tag{1}$$

where we loosely use $\circ$ as the application of transformation $g$[1]. This property implies the output of a function is left

---

[*]Corresponding author: p.a.vanderlinden@uva.nl

[1]Formally, a group element $g$ acts on a space via the group action $\rho(g)$. See Bronstein et al. [2021] for a thorough treatment.

*Accepted for the 8th Workshop on Tractable Probabilistic Modeling at UAI* (TPM 2025).

unchanged when the input is transformed by $G$. Similarly, a function is said to be *equivariant* if it satisfies

$$g \circ f(\mathbf{x}) = f(g \circ \mathbf{x}) \quad \forall g \in G, \qquad (2)$$

meaning that the output transforms in a structured, predictable way under the action of $G$. While equivariance preserves geometric structure through the transformation, invariance discards it. These properties are usually enforced through architectural constraints that ensure transformation-preserving layers, *e.g.* Cohen and Welling [2016], Weiler and Cesa [2019]. However, invariance can also be approximately enforced via *data augmentation* during training, encouraging the model to produce identical outputs for transformed inputs [Lyle et al., 2020].

**Conformal prediction and calibration.** *Conformal prediction* offers a popular framework to extend a model's pointwise predictions to prediction set estimation. The approach is fully data-driven, *post-hoc*, and amenable to both classification and regression [Fontana et al., 2023]. Crucially, relying on a data exchangeability argument (*i.e.* relaxed *i.i.d.*'ness) a probabilistic coverage guarantee for unseen test samples can be provided, attaching a notion of reliability to obtained uncertainties [Shafer and Vovk, 2008]. In contrast, probabilistic *calibration* does not provide explicit guarantees, but instead captures a notion of asymptotic consistency between predicted and observed outcomes [Guo et al., 2017, Silva Filho et al., 2023]. That is, whether a model's prediction confidence $\hat{p}$ aligns with the target's true observed frequency $p$ in the data, rendering it trustworthy.

**Bayesian model selection.** Within a Bayesian context, a model $\mathcal{M}$'s ability to explain observed data $\mathcal{D}$ can be evaluated via the *marginal likelihood*, formally denoted as

$$p(\mathcal{D}|\mathcal{M}) = \int p(\mathcal{D}|\boldsymbol{\theta}, \mathcal{M}) \, p(\boldsymbol{\theta}|\mathcal{M}) \, \mathrm{d}\boldsymbol{\theta}. \qquad (3)$$

Also referred to as *model evidence*, the quantity averages the data likelihood $p(\mathcal{D}|\boldsymbol{\theta}, \mathcal{M})$ over the prior $p(\boldsymbol{\theta}|\mathcal{M})$ for model parameters $\boldsymbol{\theta}$, effectively quantifying data fit while penalizing overly complex models that assign low prior probability to regions of high likelihood [MacKay, 2003]. This naturally integrates *Occam's Razor* as a model selection principle, where the model with highest $p(\mathcal{D}|\mathcal{M})$ is preferred [Rasmussen and Ghahramani, 2000] and arguably generalizes more favourably [Germain et al., 2016, Lotfi et al., 2022]. As Eq. 3 is generally intractable, and in particular for large-scale neural networks, efficient and scalable approximations become necessary [Llorente et al., 2023].

## 3 UNCERTAINTY MEASURES FOR MODEL SELECTION

We next detail the particular uncertainty-based measures we employ based on the above frameworks to assess model fit.

**Conformal prediction set size.** The satisfaction of conformal coverage guarantees at some target level $1-\alpha$ (*e.g.* 90%) is met by design, thus we focus on assessing model fit by the *efficiency* of obtained prediction sets, where smaller set size indicates more informative uncertainty [Shafer and Vovk, 2008]. Given test observations $(\mathbf{x}_i, \mathbf{y}_i) \in \mathcal{D}_{test}$ and produced prediction sets $C(\mathbf{x}_i)$, the *mean set size* is simply

$$\text{Mean set size} = \frac{1}{|\mathcal{D}_{test}|} \sum_{i \in \mathcal{D}_{test}} |C(\mathbf{x}_i)|. \qquad (4)$$

We combine the models outlined in § 4 with simple top-class and residual scoring to construct the conformal prediction sets (see Appendix B), and refer to Angelopoulos et al. [2023] for an introductory work covering these mechanisms.

**Calibration error.** For classification, the *expected calibration error* (ECE) is commonly employed despite some pathologies [Guo et al., 2017, Nixon et al., 2019]. Therein confidence levels are binned and the gap to similarly binned model accuracies is measured, thus favouring a smaller ECE that better aligns model confidence and output. We additionally measure the *Brier score* [Brier, 1950], a classical probabilistic scoring rule whose decomposition expresses both calibration and efficiency properties of the model [Gneiting et al., 2007, Murphy, 1973]. For regression tasks the notion of calibration is ambiguously defined, but tends to pertain to coverage properties of estimated intervals [Kompa et al., 2021, Kuleshov et al., 2018]. In that context, conformal prediction interval size can also be framed as a calibration measure [Dheur and Taieb, 2023], which we consider here.

**Bayesian model selection via Laplace approximation.** We employ Laplace's method [MacKay, 1992] to obtain a tractable approximation of Eq. 3. Therein, a second-order Taylor expansion for the unnormalized log-posterior $p(\boldsymbol{\theta}|\mathcal{D}, \mathcal{M})$ around a local optimum $\boldsymbol{\theta}_*$ (here, the pretrained model's weights) yields the distinct terms

$$\log p(\mathcal{D}|\mathcal{M}) \approx \underbrace{\log p(\mathcal{D}|\boldsymbol{\theta}_*, \mathcal{M})}_{\text{Data fit}}$$
$$- \underbrace{\left[\tfrac{1}{2} \log |\tfrac{1}{2\pi}\mathbf{H}_*| - \log p(\boldsymbol{\theta}_*|\mathcal{M})\right]}_{\text{Model complexity}}, \qquad (5)$$

where $\mathbf{H}_* = -\nabla_{\boldsymbol{\theta}}^2 \log p(\mathcal{D}|\boldsymbol{\theta}_*, \mathcal{M}) + \delta\mathbf{I}$ is the Hessian of the negative log likelihood and $\delta$ is the precision of the isotropic Gaussian prior $p(\boldsymbol{\theta}|\mathcal{M}) = \mathcal{N}(\mathbf{0}, \delta^{-1}\mathbf{I})$. The trade-off between data fit (the log-likelihood evaluated at $\boldsymbol{\theta}_*$) and model complexity in the Bayesian sense becomes apparent. As we aim for *post-hoc* model selection, we follow the recommendation of Daxberger et al. [2021] for a *last-layer* Laplace approximation[2], wherein $\mathbf{H}_*$ is computed only over the last linear model layer. Other approximations such as KFAC or diagonal are also possible [Immer et al., 2021].

---

[2]Using https://aleximmer.com/Laplace/

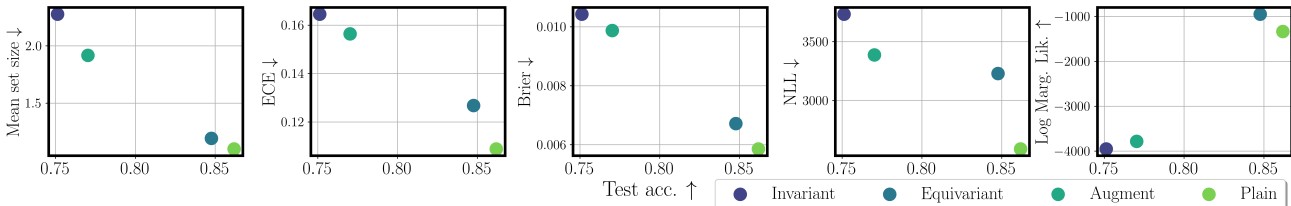

Figure 1: We visualize alignment between uncertainty-based measures (*y-axis*) and prediction accuracy (*x-axis*) on Model-Net40 test data for all four models (§ 4). 'NLL' refers to the *negative log-likelihood* of the model's direct softmax output (*i.e.* not using Laplace), while 'Log Marg Lik' equates Eq. 5. ($\uparrow\downarrow$) indicate the desired direction of the measure.

| Target | Model $\mathcal{M}$ | Train data $\mathcal{D}_{train}$ | | | | Test data $\mathcal{D}_{test}$ | |
| | | MAE $\downarrow$ | LogLik $\uparrow$ | Complexity $\downarrow$ | Log-MargLik $\uparrow$ | MAE $\downarrow$ | LogLik $\uparrow$ |
|---|---|---|---|---|---|---|---|
| $\mu$ | Invariant | 0.0025 | -101084 | 767 | -101851 | 0.0204 | 22064 |
| | Equivariant | 0.0083 | -101091 | 723 | **-101814** | **0.0145** | **22940** |
| | Augment | 0.0048 | -101086 | 799 | -101886 | 0.0254 | 20826 |
| | Plain | 0.0038 | -101086 | 798 | -101884 | 0.0296 | 19622 |
| $\alpha$ | Invariant | 0.0102 | -101097 | 1530 | **-102628** | 0.0613 | 768 |
| | Equivariant | 0.0290 | -101176 | 1515 | -102691 | **0.0522** | **9014** |
| | Augment | 0.0153 | -101112 | 1521 | -102633 | 0.0679 | -3732 |
| | Plain | 0.0106 | -101100 | 1564 | -102664 | 0.0888 | -19273 |
| $\varepsilon_{HOMO}$ | Invariant | 0.2540 | -101083 | 1211 | -102295 | 23.4848 | 20586 |
| | Equivariant | 2.9681 | -101084 | 1243 | -102327 | **21.3705** | **20989** |
| | Augment | 0.7900 | -101083 | 1149 | -102233 | 27.4825 | 19461 |
| | Plain | 0.2288 | -101083 | 1148 | **-102231** | 33.6994 | 17578 |
| $\varepsilon_{LUMO}$ | Invariant | 0.9781 | -101083 | 416 | -101500 | 20.2211 | 21650 |
| | Equivariant | 4.3102 | -101085 | 465 | -101550 | **19.3362** | **22028** |
| | Augment | 1.3789 | -101083 | 414 | -101498 | 21.6268 | 21417 |
| | Plain | 1.0273 | -101083 | 361 | **-101445** | 24.2827 | 20628 |
| $C_\nu$ | Invariant | 0.0134 | -101112 | 1345 | -102457 | 0.0298 | 19330 |
| | Equivariant | 0.0180 | -101121 | 1326 | -102447 | **0.0275** | **20017** |
| | Augment | 0.0107 | -101104 | 1334 | **-102438** | 0.0331 | 18098 |
| | Plain | 0.0066 | -101094 | 1357 | -102451 | 0.0394 | 15812 |

Table 1: We tabularize results for all four models (§ 4) and five regression targets on QM9 train and test data. We report predictive error (via the *mean absolute error*, MAE) and the Laplace-based terms in Eq. 5: data fit via the log-likelihood (LogLik), Bayesian model complexity, and the overall log-marginal likelihood (Log-MargLik). ($\uparrow\downarrow$) indicate the desired direction of the measure, and we highlight the **preferred model** according to different selection criteria.

## 4    EXPERIMENTAL RESULTS

We next outline our experimental design and results, and defer further implementation details to Appendix B. We consider two prediction tasks and datasets: classification on ModelNet40 [Wu et al., 2015] and regression on QM9 [Ramakrishnan et al., 2014]. ModelNet40 consists of $\sim$ 12 000 rotationally aligned objects across 40 classes and hence does *not* strictly require $SO(3)$ invariance. QM9 contains $\sim$ 130 000 molecular point clouds with several scalar targets, of which we consider $\mu, \alpha, \varepsilon_{HOMO}, \varepsilon_{LUMO}$ and $C_\nu$ (separate models are trained for each target). Due to arbitrary molecular orientations, invariance to the $SO(3)$ group is generally considered necessary here.

**Model choice.** We evaluate four variations of the message-passing architecture PONITA [Bekkers et al., 2024], which are implemented in Vadgama et al. [2025] as Rapidash.

The Rapidash model is well-suited for our study as it enables explicit control over the equivariance constraints employed within layers, and else maintains the same architecture. We consider the following choices:

1. **Invariant**: constrained via invariant message passing layers;
2. **Equivariant**: constrained via equivariant message passing layers;
3. **Augment**: same as **Plain** but trained with $SO(3)$ data augmentations;
4. **Plain**: fully expressive and unconstrained, without $SO(3)$ equivariance.

In terms of geometric (*i.e.* function-fitting) expressivity, Models 3 and 4 are most expressive, while Model 1 is the most constrained. Model 2 leverages geometric structure for richer representations despite (intermediate) constraints.

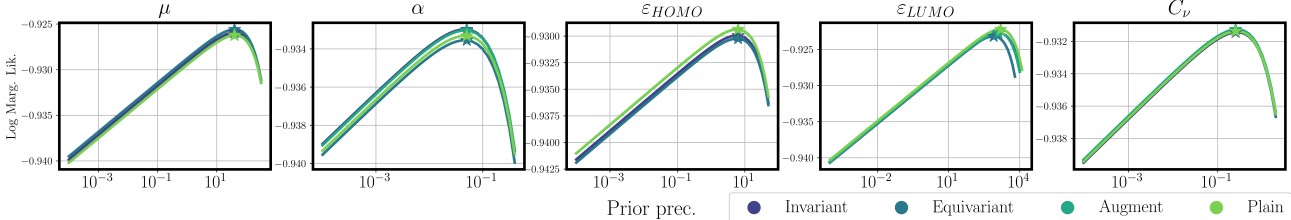

Figure 2: We optimize the Laplace approximation's prior precision parameter $\delta$ in a *post-hoc* fashion via grid search using the marginal likelihood (Eq. 5). We visualize the grid's values and optimum ($\star$) for every QM9 regression target and model.

**ModelNet40 classification results.** We visualize our results across various uncertainty-based measures in Fig. 1. Conformal and calibration measures closely align with prediction accuracy, and similarly suggest the unconstrained model (Plain) as the best fit for rotationally aligned data. In contrast, the Log-MargLik slightly favours the equivariant model despite its lower accuracy (*i.e.* presumed data fit), suggesting a slight effect of the model complexity term (discussed further in § 5). In contrast, for $SO(3)$-rotated ModelNet40 (Fig. 3) the geometric inductive bias becomes crucial, and all measures correctly identify the equivariant model as the preferred choice.

**QM9 regression results.** Tab. 1 reports results for prediction error – as measured via the *mean absolute error* or MAE – and the marginal likelihood (Log-MargLik), including its individual terms on data fit (LogLik) and model complexity (Eq. 5). Note that the Log-MargLik is rightly evaluated on the train data on which the Laplace approximation is computed. Measures on test data (incl. the Laplace's test log-likelihood) clearly express a preferred model ranking in line with expectations on this $SO(3)$-affected task, that is Equivariant > Invariant > Augment > Plain. Conformal results are given in Fig. 4 and also align with the MAE. In contrast, the Log-MargLik fails to capture this trend and varies preferences across targets. Despite slight variations in train MAE the log-likelihoods are extremely similar and entirely dominate the complexity term, resulting in indistinguishable marginal likelihoods. Given the lack of coherence in obtained model complexities, we hypothesize that our Laplace-based approximation fails to properly account for *geometric* expressivity as purportedly captured in the model's feature structures. We discuss this effect next.

## 5 DISCUSSION & OUTLOOK

The marginal likelihood can also be leveraged in an "empirical Bayes" fashion to optimize hyperparameters of the Laplace approximation (even *post-hoc*), such as the prior precision $\delta$ [Immer et al., 2021]. Relating this step to a model's feature expressivity, one might expect the prior precision of a more expressive model to be tuned higher as an intrinsic guard against data *over*-fit [Bishop, 2006], and

result in a larger complexity term in Eq. 5[3]. Nevertheless, our investigation on this relation in Fig. 2 finds the optimal value to be almost identical across all models. We interpret this as supporting evidence that the employed last-layer Laplace, being a relatively crude approximation of the full marginal likelihood, may lack the sensitivity to distinguish between models with different geometric properties based solely on their last-layer representations. This aligns with prior observations on the limitation of last-layer approximations [Ober et al., 2021, Schwöbel et al., 2022]. Such geometry-induced structural differences clearly exist, as exemplified by Moskalev et al. [2023] (cf. their Fig. 2) who demonstrate that data-augmented models map transformed inputs to close but distinct locations in the representation space, whereas strict invariance ensures a mapping to singular representations by design.

**Perspectives on a Geometric Occam's Razor.** How to appropriately integrate geometric inductive biases into such a Bayesian framework in general fashion remains a somewhat open challenge. Several recent works have blended the two by framing geometric constraints as *learnable* parameters under a marginal likelihood objective. While promising, these approaches remain restricted in their generality to particular transformations (*e.g.*, the required degree of augmentation [Immer et al., 2022]) or model structures (*e.g.*, formulated as a Gaussian Process kernel [van der Wilk et al., 2018, Schwöbel et al., 2022]). Perhaps more generally, the integration of such constraints as regularizers with a possible Bayesian prior interpretation [Finzi et al., 2021, Kim et al., 2023a] or more explicit distributional correspondences [Bloem-Reddy et al., 2020, Kim et al., 2023b] could prove fruitful. We touch upon other related works in Appendix A.

**Conclusion.** We explore the use of uncertainty-based measures to guide model selection among pretrained equivariant architectures. Conformal and calibration measures, while well aligned with predictive performance, offer limited insights into the underlying model fit. Bayesian model selection via the marginal likelihood shows partial promise, but requires a more thorough treatment or integration of geometric priors to enable *post-hoc* equivariant model selection.

---

[3]Such trends do implicitly assume that additional expressivity is not entirely absorbed in the data fit term.

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

# On Equivariant Model Selection through the Lens of Uncertainty
## — Supplementary Material —

## A OTHER RELATED WORK

Bayesian (and other) approaches to uncertainty quantification have been recently applied to molecular point cloud data, often using domain-specific desiderata to assess reliability [Lamb and Paige, 2020, Soleimany et al., 2021, Wollschläger et al., 2023], as well as in molecule and drug design [Mervin et al., 2021, Chen and Li, 2025]. In 3D vision, uncertainty methods have been applied to more general point cloud segmentation and classification [Petschnigg et al., 2021, Cortinhal et al., 2020]. Finally, several works leverage geometry or equivariance properties to inform conformal procedures [Kaur et al., 2022, van der Linden et al., 2025]. These approaches leverage uncertainty primarily for predictive purposes, not for general model selection.

## B IMPLEMENTATION DETAILS

| | Parameter counts | | | |
| | ModelNet40 | | QM9 | |
| **Model** | Feature extractor | Last layer | Feature extractor | Last layer |
|---|---|---|---|---|
| Invariant | 1.664.896 | | 4.638.208 | |
| Equivariant | 1.994.752 | | 5.131.904 | |
| Augment | 1.669.504 | 5160 | 4.642.816 | 257 |
| Plain | 1.669.504 | | 4.642.816 | |

Table 2: Model sizes for the feature extractors (*i.e.*, model architecture *without* last layer) and their last layers. All models share the same penultimate feature dimension within each dataset, and hence have the same last layer size.

**Training details.** Tab. 2 shows model sizes in terms of parameter counts for all models. For both ModelNet40 and QM9 we follow the architecture-specific hyperparameter configurations described in Vadgama et al. [2025]. For ModelNet40, models are trained for 250 epochs with a final layer dimension of 128. We use a 7375/2468/2468 train/validation/test split, sampling 1024 points per object. All point clouds are spatially centered and normalized, and training samples are further augmented with small random shifts drawn from $\mathcal{U}([0.0, 0.1])$. For QM9, models are trained for 300 epochs with a final layer dimension of 256. During training, node coordinates are normalized using a global scale and shift computed from the training set. We use a 110000/10000/10831 train/validation/test split.

**Data augmentation.** All feature extractors except **Plain** are pretrained using $SO(3)$ data augmentation. We treat the resulting rotational equivariance or invariance properties as implicit inductive biases learned by the feature extractor. *No data augmentation* is used during fitting of the Laplace approximation and marginal likelihood computation on the training data. This aligns with a Bayesian perspective that the marginal likelihood should only be evaluated on real observations, not augmentations [van der Wilk et al., 2018, Ober et al., 2021].

**Conformal prediction sets.** In both experiments we consider standard split conformal prediction leveraging a hold-out calibration set for the conformal procedure, and evaluate on test data [Angelopoulos et al., 2023]. We randomly resample calibration and test data splits 100 times to compute mean prediction set sizes (for classification) and interval widths (for regression) following Eq. 4. Target coverage is set to $(1 - \alpha) = 0.9$, *i.e.* a desired coverage rate of 90%. We omit reporting coverage since the rate is equally satisfied across all models (by design). For classification we use a simple thresholding nonconformity score which ranks predictions based on the model's confidence in the true class label, following Sadinle et al. [2019]. For regression we adopt the absolute residual between predicted and true target as a simple nonconformity score, following Lei et al. [2018].

# C  ADDITIONAL EXPERIMENTAL RESULTS

**Rotated ModelNet40 classification results.**   Our experimental design for ModelNet40 is additionally employed for the rotated data setting, wherein train, validation, and test data are randomly transformed by elements from the $SO(3)$ group. Results are shown in Fig. 3. We omit the unconstrained model (**Plain**) as it performs extremely poorly due to lack of $SO(3)$ generalization and is therefore uninformative. Results for the other methods highlight the same alignment observed in Fig. 1.

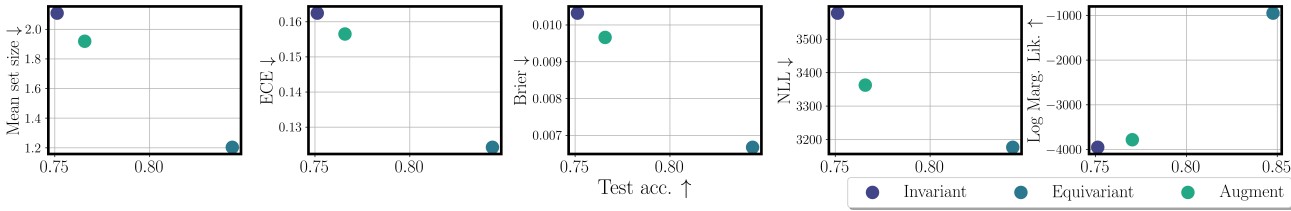

Figure 3: We visualize alignment between uncertainty-based measures (*y-axis*) and prediction accuracy (*x-axis*) on **rotated** ModelNet40 test data for three models. 'NLL' refers to the *negative log-likelihood* of the model's direct softmax output (*i.e.* not using Laplace), while 'Log Marg Lik' equates Eq. 5. ($\uparrow\downarrow$) indicate the desired direction of the measure.

**Conformal prediction set size for QM9 regression.**   We display the conformal mean set size (*i.e.* interval widths) for considered QM9 regression targets in Fig. 4. Consistent with the trends discussed in § 4 and Fig. 3 the measure closely aligns with predictive performance.

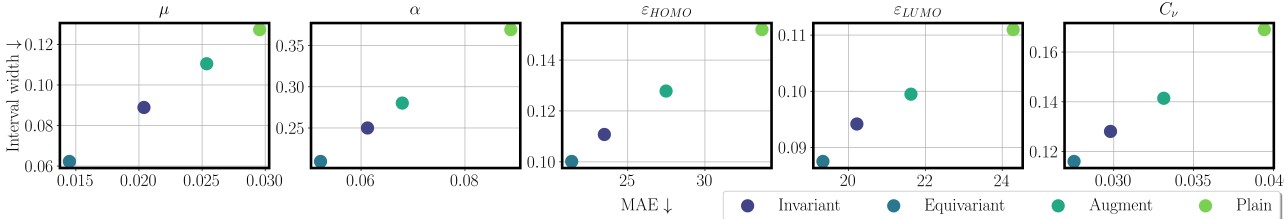

Figure 4: We visualize alignment between conformal mean set sizes or interval widths (*y-axis*) and prediction error (*x-axis*) for all four models (§ 4) and five regression targets on QM9. ($\uparrow\downarrow$) indicate the desired direction of the measure.