# OpenReview forum: "On Equivariant Model Selection through the Lens of Uncertainty"
_auai.org/UAI/2025/Workshop/TPM — TPM 2025_

### Official Review · Reviewer_Zj3j · 2025-06-15
**Interesting, but is it relevant to TPM?**

**Rating:** 2

**Review:**

The paper studies Bayesian model selection under symmetric data and claims a mismatch between
Bayesian model evidence and predictive performance under this geometric constraint. They provide
experiments with Bayesian model selection where the posterior is approximated with the Laplace
method.

The paper is well written, and although symmetry-aware model selection and conformal prediction
is definitely of interest to the TPM community, the paper itself appears to only be tangentially
related to TPM.

Furthermore, the authors claim that Bayesian model evidence does not consistently align with
predictive performance, however the paper only evaluates on two datasets, in which one (ModelNet40)
seems to show some alignment with performance, while the other (QM9) does not. The claim would be
more convincing if at least two more datasets were evaluated, especially since strong assumptions
are being made (Laplace approximation, specific architectures and selection over specific layers).

In summary, although the paper is well written and has interesting insights into Bayesian model
selection in symmetry-rich data, TPM does not seem to be the best fit. I must add however that I am
far from an expert in model selection and symmetry representation, and my evaluation of this paper
could be wrong. Thus, I would prefer to defer judgement of acceptance to other reviewers.

Below are some pointwise comments.

Section 3, Conformal prediction set size: The first sentence seems to be missing a few words.

What is Log-MargLik marginalizing over? Perhaps it is obvious within the community, but as an
outsider I am confused what exactly is being marginalized. Further, unless the marginalization is
over the models, how is it tractable?

---

### Official Review · Reviewer_2Yc4 · 2025-06-16
**interesting message**

**Rating:** 3

**Review:**

# Summary

The authors evaluate different model selection metrics in the context of equivariant model selection.  Specifically, they compare 1) a Bayesian-inspired metric, the marginal likelihood; 2) a conformal prediction metric, the mean (prediction) set size; 3) two calibration metrics, the expected calibration error and the Briar score. What they assess is the ability of these metrics to recognize the best performing model in terms of prediction accuracy (actually, test MAE and log-likelihood). The key take away is that the marginal likelihood seems to misbehave in their experiments.

# Strengths

- Well articulated, the text is crisp and easy to follow.

- The motivation is reasonable.

- The experiments are set up appropriately.

- The results are potentially useful: they might prevent future improper model selection of invariant/equivariant models.

# Weaknesses

- The experiments are quite small scale, in that they focus on two data sets.  My understanding is that this is a sort of WIP report, so this is fine.  The message is interesting anyway.

- It would be best if the robustness of the results were statistically validated.  This will probably be the case in future versions of the paper.

- Significance is modest: the target models (invariant/equivariant networks) are relatively niche.  The message is interesting anyway for researchers in that field.